# $Coder\text{-}R^3$: Recognize, Review and Repair Defective Code with Finetuned LLMs in Practice

## Abstract

Large language models have emerged as powerful tools for software engineering tasks, demonstrating particular promise in code review activities. However, existing research on code review LLMs typically decomposes the review process into discrete subtasks, collecting data and fine-tuning separate models for each individual component. This fragmented approach overlooks the synergistic relationships between different tasks, necessitates multiple models with complex multi-stage invocations, and consequently exhibits limited practical applicability in real-world deployment scenarios. In this work, we advance beyond previous code review research by proposing a unified and comprehensive code review problem modeling approach. We focus on the complete code review process, including **R**ecognize, **R**eview, and **R**epair defective code fragments consecutively, and propose $Coder\text{-}R^3$, an approach that enables a single LLM to handle all code review-related subtasks uniformly. Additionally, we establish a practically feasible closed-loop iterative process for industrial scenarios, encompassing data construction, model evaluation, and operational feedback integration. We rigorously evaluate the effectiveness of various strategies, including input context selection, output format, and training methodologies. $Coder\text{-}R^3$ achieves state-of-the-art performance on the CodeReviewer benchmark, and demonstrates superior effectiveness in real enterprise scenarios. Our work provides valuable insights for enterprises seeking to leverage LLMs to enhance code review efficiency.

## 1 Introduction

The code review activity (Li et al., 2022; Tufano et al., 2022; Lu et al., 2023a; Huq et al., 2022) is an integral component in modern software engineering workflow. Having been widely adopted across development platforms like GitHub, the code review process significantly contributes to software quality assurance and long-term maintainability. However, code review has always been a difficult and time-consuming activity both for reviewers and developers. The previous research (Bosu & Carver, 2013) show that developers spend $6.4$ hours on average weekly reviewing other developers' code. With the advance of information technology, software systems have become increasingly complex, making the code review process more challenging.

To address this challenge, recent research has increasingly focused on automating code review tasks (Tufano et al., 2024). Large language models (LLMs) (Radford et al., 2018; 2019; Brown et al., 2020; Achiam et al., 2023), with their strong linguistic capabilities, have shown considerable potential in handling various programming-related tasks. In an effort to alleviate the workload of software developers, many studies have explored the application of LLMs to automate code review processes (Hossain et al., 2024; Caumartin et al., 2025; Xue et al., 2024), achieving notable advancements in this area. Generally, the automated code review process is divided into three subtasks, including code change quality estimation, review comment generation, and code refinement (Li et al., 2022). However, due to the limitations of methodology and data, plenty of studies still focus on a single subtask (Dai et al., 2025; Sobania et al., 2023; Lopes et al., 2024; Cai et al., 2023), such as review comment generation or code refinement, resulting in a lack of overall code review capability. Some other studies work on the whole process, but perform these subtasks by individual models (Guo et al., 2024; Begolli et al., 2025), making the whole process complex and computationally intensive. Additionally, code review data, while relatively easy to collect, often suffers from

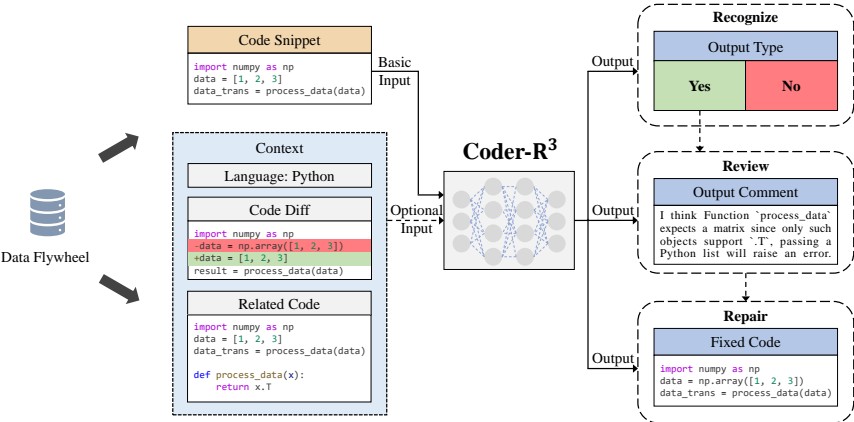

Figure 1: **The framework of our proposed Coder-R³.** Data are collected through a data flywheel mechanism. The code snippet is the code that is being reviewed. The context includes additional information helps the model to understand the code, such as programming language, code diff, and related code. Taking the code snippet and context as input, the $Coder\text{-}R^3$ model perform the Recognize, Review and Repair tasks in a unified manner.

inconsistent quality, which in turn constrains the real-world deployment of models. Overall, the current automated code review is not practical enough.

In this work, we focus on the comprehensive code review task, *i.e.* given a static code fragment, the model firstly recognizes the defect in the code, then writes a review explaining the defect and generates repaired code if the code has defection. This is a unified form of the three code review subtasks, emphasizing the connection among them. We train a single LLM to accomplish the comprehensive task, which is far more practical and efficient in real-world scenarios. Specifically, we introduce $Coder\text{-}R^3$, an approach to finetune LLMs in practical code review scenarios. As shown in Fig. 1, we fine-tune LLMs via Supervised Fine-Tuning (SFT) with uniformly formatted datasets from data flywheel, improving the overall ability for code review. Through extensive experimentation, we investigated the impact of various input configurations, output representations, and training strategies on the code review task, ultimately identifying the optimal approach. Our method achieves state-of-the-art results on the CodeReviewer benchmark.

In terms of practical application, we integrate the model into the company's actual code review system and establish a data flywheel mechanism. Within this system, the model generates review suggestions, which are then validated and annotated by human reviewers. The resulting data is systematically collected, filtered, and preprocessed to form a high-quality code review dataset. By iteratively training on this continuously updated dataset and redeploying the improved model, we create a closed-loop pipeline for continuous enhancement. Our experiments validate the method's strong effectiveness on company internal data and demonstrate its promising performance in real-world deployment.

Overall, our work is distinguished by the following aspects: (1) We unify the modeling of the code review task and propose a method $Coder\text{-}R^3$ designed for completing the whole code review process with a single LLM. (2) We perform extensive experiments to evaluate the effectiveness of various strategies, demonstrating the practicality and effectiveness of our method. (3) We deploy an iterative, data-centric approach that combines public datasets with real-world scenarios to continuously improve model performance. (4) Our approach achieves state-of-the-art results on the Codereviewer dataset, and also reveals strong performance in real-world scenarios.

## 2 RELATED WORK

CodeReviewer (Li et al., 2022) is a pioneering work that divides the code review process into three sub-tasks: code change quality estimation, review comment generation and code refinement. It

introduces a large-scale dataset for these tasks and employs pre-training and fine-tuning strategies to solve each task effectively.

Recent research has addressed code review sub-tasks individually. For example, CRScore++ (Kapadnis et al., 2025) integrates reinforcement learning with analysis tools and LLM feedback to optimize review comment generation, while ThinkRepair (Yin et al., 2024) uses thought-chain knowledge bases to improve vulnerability reasoning and code repair. However, these approaches focus on single tasks and lack comprehensive review capabilities. Some newer studies explore connections between tasks. DISCOREV (Ben Sghaier & Sahraoui, 2024) distills repair models to train review models, and CORAL (Sghaier et al., 2025a) enhances this with reinforcement learning and diversified feedback. Although these methods show promising results, they still rely on separate models for different tasks, increasing complexity and resource requirements. CFT (Kumar & Chimalakonda, 2024) attempts to train LLM using federated multi-task learning, but gains unsatisfactory results.

Additionally, existing code review datasets have several limitations. The CodeReviewer dataset, for instance, collects data for its three tasks separately from inconsistent sources, leading to informational gaps that complicate unified processing. Moreover, as it was crawled from GitHub without strict filtering, the dataset contains a significant amount of low-quality text. A recent study, CuREV (Sghaier et al., 2025b), used LLMs to clean the CodeReviewer data and found a large volume of irrelevant, unclear, or verbose content, underscoring the dataset's inherent weaknesses.

With respect to the context, most code review studies have overlooked its significance, relying solely on code snippets or code changes as model input. A recent research (Guo et al., 2025) demonstrates that inputting the review line number improves code repairing significantly, but it lacks practicability in real-world scenarios. Compared with code review tasks, the SWE-Bench (Jimenez et al., 2024) task emphasizes code context, as it requires inspecting vulnerabilities across the entire repository. These two types of tasks share certain similarities, suggesting that code review could also benefit from contextual information.

## 3 APPROACH

### 3.1 TASK FORMULATION

We formulate the code review task as: leveraging the relevant context, perform categorization (**R**ecognize), explanation (**R**eview), and fixing (**R**epair) for the input code snippet. We will develop a unified model capable of performing these three tasks simultaneously, addressing these three highly interrelated tasks in a single framework.

As shown in Fig. 1, the input of this task is $X = \langle code\_snippet, context \rangle$. Here, the code snippet ($code\_snippet$) is extracted from the modified location, and concatenated with the preceding and succeeding $N$ lines. Compared with using code changes as input, adopting code snippets is a more general approach. The relevant context ($context$) may include various forms such as function calls, documentation, or code diffs. When the context is empty, the model is expected to review only the $code\_snippet$ itself. The output is represented as $Y = \langle type, comment, fix\_code \rangle$. Each field is explained as follows: **type**: This indicates the classification categories. Similar to CodeReviewer, 'yes' signifies that a review is needed, while 'no' means no review is necessary. **comment**: For defective code, the model should generate comments that explain the issue and propose possible solutions. **fix_code**: For defective code, the output should include the corrected version of the code snippet based on the comment.

In our experiments, we employed various input-output configurations and training strategies to explore best practices for the code review task. For the input, the contextual information of the code included one or more of the following elements: programming language, code change diff, and extracted contextual code segments. For the output of the repair task, the result could be represented either as a modified code snippet or in the form of a diff. Regarding the modeling approach, one method involved a single model performing all three tasks simultaneously, directly generating a structured output containing $\langle type, comment, fix\_code \rangle$. This integrated approach offered faster inference but posed greater learning challenges. Alternatively, a more conventional automatic code review pipeline was implemented by invoking the model three separate times, each dedicated to one

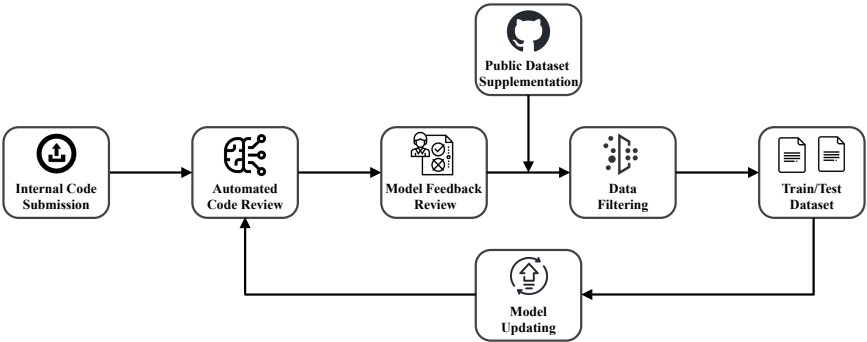

Figure 2: **Data flywheel.** Within the data flywheel, company employees submit code, which is automatically reviewed by the model. The review outcomes, combined with publicly available code data, undergo human verification and systematic data filtering. The resulting high-quality data are then used to iteratively retrain the model, establishing a self-reinforcing cycle that progressively enhances both model performance and dataset quality.

specific task. Both approaches utilize only a single model. We refer to these two methodologies as joint training and hybrid training, respectively.

## 3.2 CODE CONTEXT CONSTRUCTION

**Language.** The programming language of the code has a significant impact on defect analysis. For the majority of the collected data, language labels are already available. For the small portion of data without such annotations, we employed the open-source tool GuessLang [1] along with the LLMs Qwen2.5-Coder-32B (Hui et al., 2024) and Qwen3-Coder-30B-A3B (Team, 2025) to infer the programming language. The final label for each instance was determined through majority voting among the three systems.

**Code diff.** The code diff, which refers to the modifications made to the code compared to its previous version, serves as additional information to highlight potential defect regions for the model. During dataset construction, the code changes submitted by users are extracted and utilized as the code diff.

**Related code.** The functions invoked within a code snippet may not be included in the given snippet. Consequently, incorporating related code can provide complementary contextual information, thereby enabling a more comprehensive analysis of potential code defects. We parse source code files into abstract syntax trees using Tree-sitter [2] and adopt functions as the slicing unit. For each function snippet, we additionally capture its internal call relations, which are the set of functions invoked within its body. This approach allows each snippet to be represented not only by its own implementation but also by the contextual information regarding dependencies on other functions.

## 3.3 DATA FLYWHEEL

Manual annotation of code review data demands expert-level programmers and remains highly time-consuming. Even with model-assisted pre-annotation, each instance still takes an average of $3 \sim 5$ minutes. In practice, we designed a relatively low-cost strategy as shown in Fig. 2: deploying the system internally to collect human feedback, cleaning publicly available datasets, and introducing a data filtering process to ensure quality.

**Internal Data Collection.** We collect internal data by integrating open-source models (e.g., Qwen2.5-Coder) and commercial APIs (e.g., GPT-5) into a company's code management platform. When an engineer submits a merge request, the system reviews the code snippets and contextual information. Approved code is integrated directly, while flagged segments receive model-generated annotations for issues and fixes. Reviewers adjust these annotations and provide feedback by accepting or rejecting suggestions. Accepted recommendations are paired with the final merged code to create high-quality labeled data, enhancing the training dataset and improving the model iteratively.

---

[1]https://github.com/yoeo/guesslang
[2]https://github.com/tree-sitter/tree-sitter

**Public Dataset Supplementation.** Additional data are collected from public datasets and GitHub, then mapped to a standardized format. These data are used for training models, increasing the size of the dataset and enhancing the generalizability of the model.

**Data Filtering.** For the collected data, we conducted multiple rounds of human-AI collaborative screening to ensure high quality. An initial consistency check was performed using multi-round model voting. Data with low consensus was subsequently reviewed and corrected by human annotators. We developed a lightweight annotation system that allows code experts to examine and modify defect categories, locations, review comments, and repair suggestions. In the first round of annotation, we engaged 21 code experts to inspect and label 968 instances.

**Iterative Process.** The process follows a regular cyclic schedule, typically every one to two weeks, during which new data is collected to train an updated model. This model is then deployed to the production environment, facilitating a subsequent round of data collection and enabling continuous and scheduled model iteration.

## 3.4 MODEL TRAINING AND EVALUATION

We utilize Supervised Fine-Tuning to train $Coder\text{-}R^3$. The base pre-trained model is Qwen2.5-Coder-7B (Hui et al., 2024). The training of the model $Coder\text{-}R^3_{SFT}$ requires the raw dataset to be in the following format: $D_{SFT} = (X_i = \langle code\_snippet, context \rangle, Y_i = \langle type, comment, fix\_code \rangle)$. Instruction fine-tuning data in the form of question–answer (QA) pairs is generated through embedding $D_{SFT}$ into the corresponding template $Prompt_{SFT}$. $Q$ denotes the question serving as the input prompt, while $A = (A_1, A_2, \ldots, A_T)$ represents the reference answer consisting of a sequence of tokens. This data is then used for SFT training, in which tokens of the model's responses are compared with the ground-truth answers using cross-entropy loss to optimize the model.

$$\mathcal{L}_{\text{SFT}}(\theta) = -\sum_{t=1}^{T} \log P_\theta(A_t \mid Q, A_{<t}) \tag{1}$$

At each time step $t$, the model with parameters $\theta$ predicts the probability of the next token $A_t$ conditioned on the question $Q$ and the previously generated tokens $A_{<t}$. The objective is to maximize the likelihood of the reference answer sequence.

## 4 EXPERIMENTS ON PUBLIC DATASET

### 4.1 THE CODEREVIEWER DATASET

We choose Codereviewer (Li et al., 2022) dataset from several public datasets because it's the only large-scale multilingual public dataset possessing data on the three code review tasks. Despite some quality issues, Codereviewer dataset is a commonly used dataset in code review tasks. Other datasets like Tufano (Tufano et al., 2022) and CodeXGLUE (Lu et al., 2021) lack some information or collect data for each task from completely different sources, which are unsuitable for our code review setting. We performed a series of preprocessing steps on the dataset to align it with the specific requirements of our task setup. The details of the dataset and the preprocessing procedure are described in Appendix A.1.

### 4.2 EXPERIMENTAL SETTING

We implemented our approach using the LLaMA-Factory (Zheng et al., 2024) framework, employing LoRA (Hu et al., 2022) technique during SFT to enable efficient fine-tuning. The training is finished on four H100 80G GPUs with the learning rate of $2e-5$, and the batch size for each GPU is 1 because of the long context.

### 4.3 EVALUATION METRICS

We evaluate the three code review tasks following Codereviewer's setting.

Table 1: **Ablations on the Codereviewer dataset.** The configuration utilizing language identifiers and code diffs as input context, code snippets as output, and a hybrid training strategy yielded the best overall performance.

| Context | | Task | Output | Recognize | | | | Review | | Repair | | | |
| Language | Diff | | | Precision | Recall | F1-Score | Accuracy | Accuracy | BLEU | Accuracy | BLEU-review | BLEU-code | EM |
|---|---|---|---|---|---|---|---|---|---|---|---|---|---|
| ✗ | ✗ | Joint | Snippet | 56.73 | 59.28 | 57.98 | 57.14 | 55.36 | 5.47 | 79.15 | 8.07 | 85.42 | 9.27 |
| ✓ | ✗ | Joint | Snippet | 57.00 | 58.25 | 57.62 | 57.21 | 54.52 | 5.32 | 77.65 | 8.25 | 85.43 | 9.51 |
| ✓ | ✓ | Joint | Snippet | **99.90** | 13.80 | 24.25 | 56.94 | 4.57 | 5.37 | 99.72 | 9.76 | 85.53 | 16.12 |
| ✗ | ✗ | Recognize | Snippet | 75.34 | 61.39 | 67.65 | 70.66 | - | - | - | - | - | - |
| ✗ | ✗ | Review | Snippet | - | - | - | - | - | 5.39 | - | - | - | - |
| ✗ | ✗ | Repair | Snippet | - | - | - | - | - | - | - | - | 85.58 | 41.05 |
| ✗ | ✗ | Hybrid | Snippet | 79.33 | 58.77 | 67.52 | 71.74 | - | 5.33 | - | - | 85.65 | 40.48 |
| ✓ | ✗ | Hybrid | Snippet | 76.25 | 63.47 | 69.28 | 71.86 | - | 5.37 | - | - | 85.58 | 41.15 |
| ✓ | ✓ | Hybrid | Snippet | 79.68 | **71.60** | **75.43** | **76.68** | - | 5.61 | - | - | **86.97** | **45.47** |
| ✓ | ✓ | Hybrid | Diff | 81.14 | 60.85 | 69.55 | 73.36 | - | **5.68** | - | - | 85.54 | 40.89 |

**Recognize Task.** This is a binary classification task aimed at determining whether a given code snippet requires review. We use precision, recall, F1-score, and accuracy to evaluate the model's predictions. For the calculation of precision, recall, and F1-score, code snippets that need review are treated as the positive class. This helps in clearly quantifying how well the model identifies snippets that actually require attention from reviewers.

**Review Task.** For this sequence generation task, we adopt the BLEU (Bilingual Evaluation Understudy) (Papineni et al., 2002) score to assess the quality of the generated review comments.

$$BLEU = BP \cdot \exp\left(\frac{1}{N}\sum_{n=1}^{N}\log p_n\right) \tag{2}$$

$$BP = \begin{cases} 1 & \text{if } c > r \\ \exp\left(1 - \frac{r}{c}\right) & \text{if } c \leq r \end{cases} \tag{3}$$

where $p_n$ denotes the proportion of n-grams in the candidate text that also appear in the reference text. $c$ and $r$ denote the lengths of the candidate text and the reference text. $N$ is the maximum n-gram order used in the calculation, which is set to 4 in practice. And the brevity penalty $BP$ is introduced to penalize candidate texts that are too short. When computing BLEU, all whitespace is normalized to a single space and punctuation is separated from words as independent tokens, providing a consistent tokenization and spacing scheme for n-gram statistics.

**Repair Task.** To evaluate the performance of this task, we calculate the BLEU score between the generated code and the target code, as well as the exact match (EM) rate. When computing the BLEU score for code snippets, we preprocess the data by removing leading and trailing spaces from each instance and compressing consecutive spaces into a single space, consistent with the procedure in Codereviewer(Li et al., 2022) to ensure fairness. While BLEU reflects the similarity between the generated code and the target code, the EM rate is more critical here because code is highly sensitive to changes—even minor modifications can lead to compilation errors or runtime exceptions. Only when the generated code is completely identical to the target code is it considered a successful refinement.

## 4.4 ABLATION STUDY

We investigate the contribution of each variable via extensive ablations, and shown in Tab. 1.

**Modeling Approaches.** For the joint training approach, the calculation of BLEU and Exact Match (EM) scores excluded samples that were incorrectly classified as "requiring no review", considering only those correctly identified for further tasks. Across all experimental settings, joint training consistently underperformed compared to hybrid training. This may be attributed to the excessive information the model must process simultaneously in the joint setup, hindering its ability to effectively address each subtask sequentially. The results suggest that integrating all three tasks into a single model remains a challenging objective.

Table 2: **Evaluation on the Codereviewer dataset.** $Coder\text{-}R^3$ achieves state-of-the-art results on the Recognize and Repair tasks, surpassing all baseline pre-trained LLMs and most of prior works. (- indicates that the model did not perform this task or did not test this metric. * indicates the result of using DeepSeek-7B, as it matches $Coder\text{-}R^3$ in parameter scale.)

| Methods | Recognize | | | | Review | Repair | |
| --- | --- | --- | --- | --- | --- | --- | --- |
| | Precision | Recall | F1-Score | Accuracy | BLEU | BLEU | EM |
| Qwen-2.5-Coder-7B (Hui et al., 2024) | 47.05 | 13.65 | 21.17 | 49.16 | 0.63 | 74.96 | 22.38 |
| Qwen-2.5-Coder-14B (Hui et al., 2024) | 56.32 | 8.64 | 14.99 | 50.98 | 0.54 | 74.91 | 26.22 |
| Qwen-2.5-Coder-32B (Hui et al., 2024) | 54.93 | 39.87 | 46.21 | 53.59 | 0.51 | 72.69 | 26.36 |
| OpenAI GPT-5 (OpenAI, 2025) | 51.06 | 75.00 | 60.75 | 51.30 | 0.42 | 76.25 | 31.63 |
| Codereviewer (Li et al., 2022) | 78.60 | 65.63 | 71.53 | 73.89 | 5.32 | 82.61 | 30.32 |
| LLaMA-Reviewer (Lu et al., 2023b) | 60.99 | **83.50** | 70.49 | - | 5.70 | 82.27 | - |
| CFT-reg(Kumar & Chimalakonda, 2024) | 49.14 | 62.90 | 55.17 | - | 0.67 | 76.10 | - |
| DISCOREV (Ben Sghaier & Sahraoui, 2024) | - | - | - | - | 7.33 | 85.49 | - |
| CoRAL (Sghaier et al., 2025a) | - | - | - | - | **8.67** | | |
| Toggle(Hossain et al., 2024) | - | - | - | - | - | - | 25.59 |
| Intention is All You Need*(Guo et al., 2025) | - | - | - | - | - | - | **50.04** |
| $Coder\text{-}R^3$ | **79.68** | 71.60 | **75.43** | **76.68** | 5.61 | **86.97** | 45.47 |

**Input Features.** The inclusion of programming language information led to minor improvements across most metrics, whereas adding diff information yielded substantially greater gains. For example, accuracy improved by $0.12\%$ with language metadata and by $4.82\%$ with diff information. Similar trends were observed in other metrics, with the exception of the Repair task, where adding language metadata caused a very slight decrease in BLEU score ($-0.07$). These findings confirm that both types of contextual features contribute positively to model performance on this dataset.

Notably, under the joint training setup, incorporating diff inputs led to severe overfitting—precision reached $99.90\%$, while recall dropped to $13.80\%$. We hypothesize that this may be due to truncated diff segments in the CodeReviewer code refinement data, creating a discernible distribution gap between data types.

**Output Format.** We also experimented with using diff-style outputs under the best-performing training configuration, introducing line numbers of every code line into the input code and asking the model to generate a diff rather than the full repaired code. During evaluation, the generated diff was applied to the original code to recover the repaired version for assessment. However, this approach underperformed compared to directly generating the full code snippet, indicating that accurately producing diff-structured edits remains a difficult task for LLMs.

### 4.5 COMPARED TO OTHER METHODS

For the CodeReviewer dataset, we employed code snippets, programming language identifiers, and code diffs as input, with the repaired code snippet as the output. The model was trained using the hybrid training approach. The same input and output configuration was applied to the baseline models for a fair comparison. The evaluation result is shown in Tab. 2.

Our proposed $Coder\text{-}R^3$ model significantly outperformed all baseline pre-trained LLMs across all three tasks, demonstrating the effectiveness of our training methodology. It is worth noting that Qwen-2.5-Coder-7B, without any fine-tuning, achieved an accuracy below $50\%$ in the binary classification task. This suggests a potential discrepancy between the labeling conventions in the CodeReviewer dataset and common understanding, leading to the pre-trained model's performance being worse than random guessing in this specific context. Interestingly, we observed that the performance on the Review task decreased as the model size increased, with even GPT-5 delivering the lowest results. This counterintuitive finding may indicate potential limitations in the data quality.

In comparisons with prior work, our model achieved state-of-the-art results on the Recognize and Repair tasks, though a performance gap remains on the Review task. For the Recognize task, our model surpassed previous methods on all metrics except Recall, with a $2.8\%$ higher accuracy than CodeReviewer. On the Review task, both DISCOREV and CoRAL leveraged the relationship between review and repair tasks to achieve superior results. For the Repair task, our model attained

Table 3: **Ablations on the internal dataset.** The optimal performance was achieved by the model configuration that used a hybrid training approach, with code diff as input context, and generated a full code snippet as output.

| Context | | | Task | Output | Recognize | | | | Review | Repair | |
|---|---|---|---|---|---|---|---|---|---|---|---|
| Language | Diff | Related Code | | | Precision | Recall | F1-Score | Accuracy | BLEU | BLEU | EM |
| ✗ | ✗ | ✗ | Joint | Snippet | 98.69 | 23.15 | 37.51 | 70.06 | 31.04 | 56.60 | 11.58 |
| ✓ | ✗ | ✗ | Joint | Snippet | 98.32 | 22.52 | 36.65 | 69.77 | 30.83 | 55.42 | 10.88 |
| ✓ | ✓ | ✗ | Joint | Snippet | **98.71** | 23.60 | 38.09 | 70.22 | 31.63 | 56.79 | 11.68 |
| ✓ | ✓ | ✓ | Joint | Snippet | 98.30 | 22.22 | 36.25 | 69.66 | 32.43 | 54.50 | 10.68 |
| ✗ | ✗ | ✗ | Hybrid | Snippet | 75.77 | 46.74 | 57.81 | 73.52 | 32.84 | 77.53 | 18.16 |
| ✓ | ✗ | ✗ | Hybrid | Snippet | 75.03 | 46.28 | 57.25 | 73.17 | 32.28 | 77.51 | **18.70** |
| ✗ | ✓ | ✗ | Hybrid | Snippet | 77.28 | **47.96** | **59.19** | **74.33** | 33.06 | **78.01** | 18.23 |
| ✗ | ✗ | ✓ | Hybrid | Snippet | 75.55 | 47.35 | 58.21 | 73.61 | **33.25** | 77.69 | 18.36 |

the highest BLEU score, while slightly trailing behind the study by Guo et al. in EM. This minor discrepancy can be attributed to their use of an Intention Framework augmented with RAG and the inclusion of associated line numbers as input, which creates a different experimental setting and precludes a direct comparison.

## 5 CODE REVIEW IN PRACTICE

Beyond utilizing public datasets, we have concentrated on ensuring the method's effectiveness and feasibility in real-world code review environments. To achieve this, we gathered internal data from a company (with over $1,000$ software engineers), trained models based on this data, and assessed the method's performance within the company's actual code review system. The experimental settings employed are consistent with those detailed in Sec. 4.2.

### 5.1 INTERNAL DATASET CURATION

This data is sourced from real-world development processes within the company. Due to the company's rigorous programming guidelines and the fact that code contributors and reviewers tend to be more diligent and responsible compared to those in open-source projects, the data is of higher quality than that typically found in public datasets. The details of the dataset and the preprocessing procedure are described in Appendix A.2.

### 5.2 EVALUATION METRICS

The metrics of three code review tasks are consistent with the Codereviewer dataset, as described in Sec. 4.3. Additionally, we introduced real-world production metrics based on the company's code review system to evaluate the model's practicability: Online Accept Rate(OAR), Replay Precision(RP) and Replay Recall(RR). OAR means the accept rate of suggestion made by the model, while RP and RR means the classification metrics of online data. More details in Appendix B.1.

### 5.3 EXPERIMENT RESULTS

**Ablation Study**. Similar to Sec. 4.4, we conduct ablations on the internal dataset, the results are shown in Tab. 3. On the internal dataset, Joint Training again underperformed compared to Hybrid Training. The Joint Training approach resulted in a model with high precision but low recall for the classification task, although the overall accuracy did not decrease significantly. This phenomenon may be attributed to the excessive number of tokens involved in the joint training process, which could have hindered effective optimization of the tokens specifically responsible for classification.

Regarding input features, the inclusion of programming language had almost no effect on the internal dataset, suggesting that the longer code segments provided sufficient context for the model to infer the language. The addition of diff information still provide a measurable advantage. While the inclusion of related code leads to minor improvements, particularly in the review task, the substantial increase in token length made its inclusion cost-ineffective. If more concise and relevant code contexts could be extracted, it would likely yield greater benefits.

Table 4: **Evaluation on the internal dataset.** $Coder$-$R^3$ demonstrates strong efficacy on the internal dataset, achieving significant improvements compared to baseline models.

| Methods | Recognize | | | | Review | Repair | |
|---|---|---|---|---|---|---|---|
| | Precision | Recall | F1-Score | Accuracy | BLEU | BLEU | EM |
| Qwen-2.5-Coder-7B (Hui et al., 2024) | 51.56 | 16.39 | 24.88 | 61.57 | 5.11 | 75.54 | 14.48 |
| Qwen-2.5-Coder-14B (Hui et al., 2024) | 54.94 | 22.14 | 31.56 | 62.73 | 5.86 | 75.83 | 13.95 |
| Qwen-2.5-Coder-32B (Hui et al., 2024) | 42.18 | **82.29** | 55.77 | 49.34 | 6.45 | 75.79 | 14.74 |
| $Coder$-$R^3$ | **77.28** | 47.96 | **59.19** | **74.33** | **33.06** | **78.01** | **18.23** |

**Compared to Other Methods**. On the internal dataset, we adopted code snippets and code diffs as input, with the complete repaired code as the output, and trained the model using a multi-stage approach. The same input-output configuration was applied to the baseline models for comparison.

As shown in Tab. 4, our model significantly outperformed the non-fine-tuned baselines across all metrics, achieving an $11.6\%$ improvement in accuracy, a $26.61$-point increase in BLEU for the Review task, and gains of $2.18$ and $3.49$ in BLEU and EM respectively for the Repair task.

The baseline models' initial performance on our dataset was notably higher on the Recognize and Review tasks compared to their results on the CodeReviewer dataset, suggesting superior label and annotation quality in our internal data. In contrast, the overall lower scores on the Repair task may be attributed to the longer code snippets and more extensive modifications required, indicating a potentially higher difficulty level than that of the CodeReviewer data.

However, we observed that the 32B model exhibited a distinct tendency to classify samples as "requiring review", a behavior not seen in the two smaller models. This bias resulted in a high recall score but a corresponding decrease in overall accuracy. This phenomenon presents an interesting direction for further investigation.

### 5.4 Performance on Real-World Deployment

We deployed a trained Qwen2.5-Coder-32B model into the company's internal code review system. On a weekly basis, we collected data on the model's performance in real-world usage. This weekly human feedback data was then incorporated into the training data, allowing us to iteratively update the model through rolling training iterations. The actual results are shown in Fig. 3. Model performance was evaluated weekly, with 'Baseline' representing the results of the untrained baseline model on the preceding week's data. A consistent upward trend was observed across all three metrics as training progressed, demonstrating the effectiveness of our training methodology and data iteration strategy. The most significant improvement occurred after the initial fine-tuning round, with subsequent gains gradually diminishing over time before the metrics eventually approached a stable asymptote.

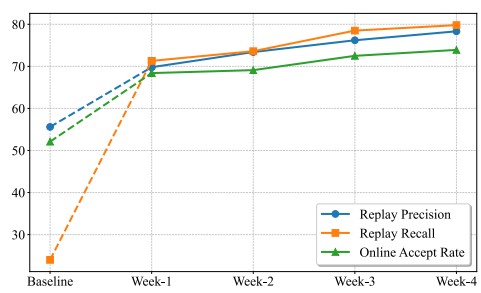

Figure 3: **Performance on real deployment.** We iteratively train and deploy $Coder$-$R^3$ by collecting human feedback data. All metrics showed a consistent increase over time, validating the efficacy of our approach.

## 6 Conclusion

In this work, we present $Coder$-$R^3$, a novel approach that enables large language models to simultaneously perform three core code review tasks: **R**ecognize, **R**eview, and **R**epair. By leveraging a data flywheel mechanism, we continuously collect internal corporate data to train our model, deploy it for practical use, and gather new feedback—forming a closed-loop iterative pipeline. Our method achieves state-of-the-art performance on the CodeReviewer benchmark and demonstrates strong effectiveness in real-world industrial scenarios, contributing significantly to the practical implementation of automated code review systems.

# 7 STATEMENTS

## 7.1 ETHICS STATEMENT

This research strictly adheres to academic ethical principles and is dedicated to promoting social well-being and human progress. We are committed to conducting research responsibly, striving to minimize potential negative consequences. Our work upholds the highest standards of scientific excellence, ensuring methodological transparency and reproducible results.

All data usage complies with ethical review approvals. The Codereviewer dataset is an open-sourced dataset with CC BY 4.0 License. The internal data we collected was obtained with the company's permission, and due to proprietary information, it will not be publicly available at this time. Whether the data will be shared in the future is contingent on company decisions.

Regarding the application of LLMs in research, we only utilize LLMs as tools for assisting and polishing writing. The core method development in this research does not involve LLMs as any important, original, or non-standard components.

## 7.2 REPRODUCIBILITY STATEMENT

We guarantee that our research is reproducible. Our data processing procedures are consistent with the descriptions in Sec. 3.3 and Sec. 5.1. The data processing and training code will be open-sourced. Our training process can be reproduced by downloading the open-source LLM Qwen-Coder-7B and fine-tuning it using the code.

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

# A  APPENDIX: DATASET DETAILS

## A.1  THE CODEREVIEWER DATASET

Codereviewer dataset is collected from GitHub repositories in nine programming languages, including C, C++, C#, Go, Java, JavaScript, PHP, Python, and Ruby. It is composed of three subsets: code change quality estimation, review generation, and code refinement. The data formats for code change quality estimation and review generation are identical, and their datasets share a common subset of data. In contrast, the code repair set utilizes a separate and distinct dataset. The training sets for these three datasets contain $266k$, $118k$, and $150k$ instances respectively, while their corresponding test sets consist of $31k$, $10k$, and $13k$ instances.

Due to inconsistent data formats in the CodeReviewer dataset, we performed a series of preprocessing steps. Firstly, we excluded incomplete entries with missing values in certain fields. Secondly, using information such as original files, modified files, and diff content provided in the dataset, we reconstructed code snippets and standardized them into a unified format. What's more, the training sets of the code change quality estimation and review generation datasets did not include programming language labels. We used the strategy described in Sec. 3.2 to label this data, which achieved an accuracy of over 97% on the test set.

For the joint training approach, each input data sample must possess labels for all three tasks to facilitate model training. We constructed a comprehensive dataset by combining samples from the code change quality estimation dataset that were flagged as 'yes' with all available samples from the code refinement dataset.

## A.2  THE INTERNAL DATASET

We utilize internal data collected according to Sec. 3.3. In the company code review system, the review model will initially provide code review labels (category, explanation, and repaired code), and engineers can accept or reject the model's suggestions and make modifications. We collect this human feedback data as high-quality labels for training and evaluating the model.

In the experiment, we collected human-annotated data over an eight-month period, focusing primarily on mainstream programming languages such as Python, Go, and C++. After preprocessing the data into a standardized format, we obtained $18,777$ instances. To mitigate the risk of data overlap between the train and test sets, we used the data from the first six months as the train set and the data from last two months as the test set. The train set and test set were divided in a ratio of approximately $4.6 : 1$. The distribution of the internal dataset by programming language is shown in Fig. 4.

Furthermore, the code segments in our internal dataset have an average length of $50.35$ lines, which is notably longer than the average of $11.83$ lines in the CodeReviewer dataset. Since longer code segments are more common in practical development, our dataset possesses greater practical relevance.

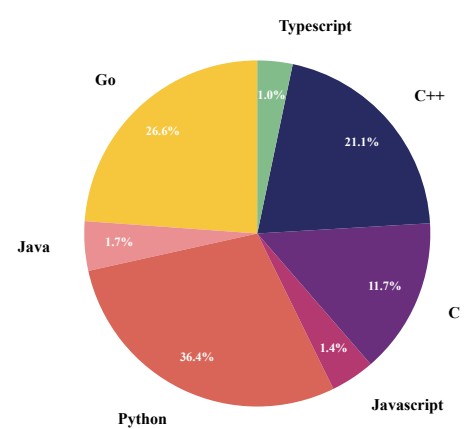

Figure 4: **Programming language distribution of the internal dataset.**

# B  METRIC DETAILS

## B.1  PRODUCTION METRICS

To evaluate the performance in real-world scenarios, we deploy our model to the code review system of the company. By tracking whether human reviewers accept or reject the model's suggestions,

these metrics offer a more realistic reflection of its practical performance, particularly for open-ended generation tasks such as comment generation and code repair.

We propose three production metrics: Online Accept Rate(OAR), Replay Precision(RP) and Replay Recall(RR). The OAR metric calculates the proportion of the model's review suggestions that are accepted by human reviewers after deployment. Let $N_{Accept}$ be the amount of accepted review instances and $N_{Reject}$ be the amount of rejected review instances during the same period of time, $OAR$ can be calculated as follows:

$$OAR = \frac{N_{Accept}}{N_{Accept} + N_{Reject}} \tag{4}$$

For RP and RR metrics, we collected human-accepted or rejected data over a recent period after the model was deployed, which, after filtering and preprocessing, was constructed into a new test set. The model was then used to predict labels on this set and calculate precision and recall rates. It's an offline method to quickly evaluate the model's online effectiveness.

