# OpenReview forum: "Coder-R3: Recognize, Review and Repair Defective Code with Finetuned LLMs in Practice"
_ICLR.cc/2026/Conference — Submitted to ICLR 2026_

### Official Review · Reviewer_UVXn · 2025-10-28

**Soundness:** 3
**Presentation:** 3
**Contribution:** 3
**Rating:** 4
**Confidence:** 4

**Summary:**

This paper proposes Coder-R3, a unified framework that enables a single large language model to perform the three main subtasks in code review — Recognize, Review, and Repair — within one cohesive system. Unlike prior work that decomposes the process into independent subtasks, Coder-R3 integrates these within a hybrid training framework that balances learning efficiency and task specialization. The authors further introduce a data flywheel mechanism, wherein real-world code review data are collected from an internal industrial system, verified by human engineers, and iteratively fed back into model training. Experiments demonstrate: State-of-the-art performance on the public CodeReviewer benchmark; Significant gains on a proprietary internal dataset with over 18k annotated instances; Sustained performance improvement in real-world deployment through continuous feedback loops.

**Strengths:**

**Unified Modeling of Code Review Tasks** Integrating recognition, comment generation, and repair tasks within a single fine-tuned LLM is novel. This unified design effectively reduces pipeline complexity and inference overhead.

**Practical Deployment and Real-World Impact** The paper extends beyond academic benchmarks by demonstrating a real-world deployment with an active feedback loop. This significantly enhances the practical value of the contribution. The proposed method achieves state-of-the-art results on the CodeReviewer benchmark and shows robust performance in an industrial environment.

**Systematic Data Design** The iterative data collection and model refinement process is well-motivated and realistic. By leveraging an in-house closed-loop system to gather high-quality annotations, the authors mitigate common issues of noise and inconsistency present in public datasets.

**Weaknesses:**

**Underperformance in the Review Task** Despite the unified design, Coder-R3 underperforms previous methods (e.g., CoRAL, GPT-5) on the Review task, suggesting that unified training may fail to fully capture linguistic and stylistic quality.

**Lack of Theoretical Analysis** The paper is largely empirical, providing limited theoretical insight or interpretability regarding why hybrid training outperforms joint training.

**Reproducibility and Generalization Concerns** Since both the dataset and deployment environment are proprietary and not publicly available, the reproducibility of industrial results is limited. Moreover, the model’s generalization ability to unseen programming languages (e.g., Rust) remains unclear.

**Technical Novelty Relatively Low** The technical novelty of the proposed approach is relatively low.

**Questions:**

- The paper claims that hybrid training significantly outperforms joint training, but the underlying reasons remain unclear. Could you provide supporting evidence such as loss curves, learning dynamics, or error-type distributions?
- How are data proportions or task weights configured across the three subtasks (Recognize / Review / Repair)? Are all validated samples included equally in the training?
- As the dataset evolves across iterations, does the model suffer from forgetting earlier knowledge? Over a longer horizon, do you observe any performance saturation or regression?
- Have you compared Coder-R3 with three separately trained task-specific models (using the same data)? How large is the performance gap between the unified and the multi-model setup?
- Compared with traditional multi-model pipelines, what are the computational costs (both training and inference)? What are the latency differences across tasks? Does the unified model offer measurable efficiency or cost advantages in production?
- How consistent is Coder-R3’s performance across programming languages? Does it exhibit any zero-shot generalization on languages unseen during training (e.g., Rust)?
- Beyond automatic metrics like BLEU, have you considered incorporating human evaluation or LLM-based assessments to better gauge comment quality?

---

> ### Author Response · Authors · 2025-12-01
>
> Thank you for your thoughtful review. We appreciate your recognition of our paper and address your concerns in detail below.
>
> ---
>
> **Question 1. Hybrid training explanation**
>
> During training process, because the outputs produced by hybrid training and joint training differ in format, the absolute loss values between the predicted outputs and reference answers are not directly comparable. **Hybrid training involves only one task of output, whereas joint training involves three tasks, resulting in inherently different loss magnitudes.** Therefore, comparing the loss curves cannot reliably indicate which training strategy is superior, so we did not use them for this purpose.
>
> We believe the reasons why hybrid training outperforms joint training are as follows.
>
>  (1) In joint training, there is a **substantial discrepancy in sequence lengths between positive and negative samples.** For samples classified as "no review needed", both the accompanying output review and repaired code are empty, whereas for samples classified as "need review", those are non-empty. The significant disparity in token counts between the answers of the two categories resulted in **suboptimal training optimization**. However, hybrid training does not suffer from this imbalance.
>
>  (2) For the **CodeReviewer dataset**, the original design provides separate datasets for the three tasks. Converting them into a unified format suitable for joint training requires **discarding a portion of the data**. As a result, within the CodeReviewer experiments, the performance gap between hybrid training and joint training tends to be larger.
>
> ---
>
> **Question 2. Data proportions**
>
> In our training process, we utilized all available data instances **equally**. Thus, the data distribution across the three sub-tasks reflects the inherent distribution of the source datasets. For the CodeReviewer benchmark, which provides distinct datasets for each task, the data ratio is approximately 2.25:1:1.27. Regarding our internal dataset, the 'Recognize' task encompasses the entire dataset, whereas the 'Review' and 'Repair' tasks are derived strictly from the subset of instances that required code review. Consequently, the data volume for the latter two tasks is smaller, resulting in an approximate ratio of 3:1:1.
>
> ---
>
> **Question 3. Continual learning**
>
> Throughout the iterative training process, we consistently integrate newly acquired data into the cumulative dataset, rather than training exclusively on incoming samples. This strategy serves to effectively **mitigate catastrophic forgetting**. We observed that after a certain number of iterations, the model's performance tends to **stabilize** and reach a plateau; however, we have not encountered any evidence of performance degradation or regression.
>
> ---
>
> **Question 4. Comparison with task-specific models**
>
> In the ablation study presented in **Table 2**, we evaluated both unified and multi-model setups. The results indicate that the two settings yield **comparable performance**; notably, the unified model exhibits a distinct advantage in the 'Recognize' task. Given that the unified approach achieves equivalent efficacy while requiring the training and deployment of only a single model, we ultimately adopted the unified training strategy as our primary methodology.
>
> ---
>
> **Question 5. Efficiency advantages**
>
> During the training phase, the total volume of training data is identical for both the unified and multi-model configurations, resulting in equivalent training computational cost. However, the multi-model approach necessitates the training of three distinct models, which significantly increases the **complexity of maintenance and updates**. In terms of inference, the unified framework requires only a single model invocation, thereby **reducing GPU memory consumption, minimizing the number of calls, and lowering end-to-end latency**. Consequently, given that the multi-model setup yielded no performance improvements while imposing greater deployment complexity, we did not deploy it in our production environment for comparative evaluation.
>
> ---
>
> **(To be continued...)**

---

> ### Author Response · Authors · 2025-12-01
>
> **(Continued)**
>
> **Question 6. Generalization**
>
> We acknowledge that transferring the model to different programming languages may indeed result in suboptimal performance. When transferring to distinct systems or languages, **fine-tuning with domain-specific data** remains essential, as coding standards and requirements vary significantly across environments. Consequently, relying solely on the model's zero-shot generalization capability is insufficient to address these specific domain constraints.
>
> However, the primary objective of this study is to validate the efficacy of Coder-R3's **training methodology** and its associated **data flywheel mechanism**. We posit that for practical deployment, the training strategy must be integrated with the data flywheel approach. It is through this synergy that the model can achieve continuous improvement and adaptation within specific domain scenarios.
>
> ---
>
> **Question 7. Evaluation metrics**
>
> We acknowledge that BLEU is not an ideal metric for evaluating code-related tasks. While recent literature has frequently highlighted this limitation, a universally accepted alternative has **yet to emerge**. For the snippet-level code review tasks addressed in this work, **we adhered to the protocols established in prior studies** by utilizing BLEU.
>
> The **production metrics** reported in this study can be interpreted as the statistical aggregation of **human judgments**, based on the premise that suggestions are typically accepted only when developers perceive them as beneficial. Since these interactions occur during actual development tasks, they capture **authentic reviewer-author dynamics** that are often difficult to replicate in user studies or LLM-base assessments. Nevertheless, we acknowledge that user studies provide a complementary and intuitive perspective on model effectiveness, and we will consider incorporating such human-centric evaluations in our future research.

---

### Official Review · Reviewer_7wdc · 2025-11-01

**Soundness:** 2
**Presentation:** 2
**Contribution:** 1
**Rating:** 4
**Confidence:** 4

**Summary:**

The paper introduces Coder-R3, a single LLM fine-tuned to handle the full code-review loop: Recognize defects, Review them with an explanation, and Repair the code. It builds a data flywheel: deploy the model in an internal review system, capture human accept/reject feedback plus public data, filter/clean it, and retrain on a regular cadence. Through ablations, the authors find a hybrid setup (invoking the same model sequentially per subtask) with diff + language context and generating full repaired snippets works best. On the CodeReviewer benchmark it reports state-of-the-art results for recognition and repair, and in a real company deployment the metrics improve steadily across iterations, suggesting practical viability.

**Strengths:**

Originality: Frames code review as a single, unified task, "Recognize → Review → Repair", handled by one fine-tuned model rather than separate models/pipelines, which prior work typically used. This integrated formulation is novel and more practical for deployment.

Quality (technical rigor): Provides concrete design choices and ablations on input context (language metadata, diffs), output format (full code vs diff), and training setups; findings are backed by measurements and useful diagnoses (e.g., overfitting when diffs are truncated).

Clarity: The paper presents a clear system picture (Figure 1) showing inputs (snippet + context like language, diff, related code) and unified outputs across the three stages, making the overall workflow easy to understand. It also explains how “related code” is sliced via Tree-sitter and call relations, which improves reader comprehension of context construction.

Significance (practical impact): Goes beyond benchmark gains to show a deployable data flywheel: internal integration to collect human accept/reject signals, public-data supplementation, quality filtering, and regular retraining, with steady real-world improvements over iterative deployments. This makes the contribution meaningful for teams aiming to productionize code-review LLMs.

**Weaknesses:**

Benchmark dependence & label noise. Most public results hinge on CodeReviewer, which the authors acknowledge has “some quality issues,” including mismatched conventions that make even strong base models underperform and diff truncation that induces overfitting under certain setups. This undermines how confidently we can read reported gains. Consider adding a second public benchmark and a manual audit of a stratified sample.

Reproducibility/data access gaps. Internal data central to the “flywheel” will not be publicly available, limiting external verification; artifact availability is promised but not yet present in the submission. Release minimally a de-identified internal evaluation set and the full preprocessing scripts.

Eval concerns. On CodeReviewer dataset eval, baselines are largely non-fine-tuned pretrained LLMs, while Coder-R3 is fine-tuned under a task-specific I/O format raising fairness concerns. And the CodeReviewer model was produced in 2022, which is certainly not SOTA for code review task anymore. You should at least compare to GPT-5 and Sonnet 4.5 with proper prompt tuning. Also BLEU score is not a good metrics for code related tasks, execution-based eval would be much more desired to validate code fix.

Limited external validity of deployment study. The real-world evaluation is within a single company’s stack and language mix, so generalization to other ecosystems is unclear. Replicate with some open-source repos and report cross-project transfer.

Human-centered quality evaluation is thin. While production metrics (OAR/RP/RR) are useful, there’s no reader study on comment helpfulness, clarity, or correctness. Add a blinded developer study with task time and fix success as outcomes.

**Questions:**

1. The paper notes that the CodeReviewer dataset has “some quality issues” and that truncated diffs may cause overfitting. Could the authors provide results on an additional public dataset or a manually audited subset of CodeReviewer to confirm that the reported gains generalize beyond this benchmark?

2. Since Coder-R3 is fine-tuned on task-specific I/O while most baselines are frozen pretrained LLMs, how do the authors ensure fairness in comparison? Would they consider releasing fine-tuned checkpoints or training scripts so that external researchers can reproduce the hybrid-training results under identical conditions?

3. The industrial deployment study focuses on automated metrics (OAR/RP/RR) but lacks direct human-judged measures of comment helpfulness or fix correctness. Could the authors add or plan a user study, e.g., developer task-time or fix-success rates, to validate whether Coder-R3 actually improves review quality in practice?

---

> ### Author Response · Authors · 2025-12-01
>
> Thank you for your thoughtful review. We appreciate your recognition of our paper and address your concerns in detail below.
>
> ---
>
> **Question 1. Benchmarks**
>
> We selected the CodeReviewer dataset primarily because of its prevalence in related works, which facilitates **direct comparative analysis**, despite its known quality issues. Other publicly available code review datasets also exhibit specific limitations, such as limited scale, restriction to a single programming language, or a lack of contextual information. To mitigate these deficiencies, we supplemented our study with experiments conducted on an **internal dataset** that has undergone rigorous **manual verification**. The purpose is to further demonstrate the effectiveness of our training method and to assess the contribution of each component through ablation studies.
>
> ---
>
> **Question 2. More information about internal dataset**
>
> In the supplementary materials, we will provide some raw data used to construct the internal dataset, as well as the full preprocessing scripts.
>
> ---
>
> **Question 3. Evaluation concerns**
>
> **Question 3.1. Compared models**
>
> In our experiments on the CodeReviewer dataset, we benchmarked against **a wide range of related works in addition to pre-trained LLMs** to ensure a fair and comprehensive comparison. Regarding powerful models such as GPT-5 and Sonnet 4.5, we acknowledge the reviewer's suggestion; however, applying prompt tuning to these closed-source models is currently not feasible. To the best of our knowledge, prompt tuning **necessitates access to model weights** to fine-tune the model. We did conduct preliminary sampling-based evaluations using GPT-5, but the resulting performance was suboptimal.
>
> **Question 3.2. BLEU score**
>
> We acknowledge that BLEU is not an ideal metric for evaluating code-related tasks. While recent literature has frequently highlighted this limitation, a universally accepted alternative has **yet to emerge**. Execution-based evaluation certainly offers a more objective assessment of code repair efficacy, but it imposes **stringent requirements** on the dataset. Currently, such evaluation capabilities are primarily available in repository-level issue-resolution benchmarks like SWE-Bench. For the snippet-level code review tasks addressed in this work, **we adhered to the protocols established in prior studies** by utilizing BLEU. Our focus is on ensuring comparability with existing baselines rather than developing novel evaluation metrics.
>
> **Question 3.3. Open-source**
>
> We will release the fine-tuned checkpoints and training scripts of Codereviewer dataset experiment.
>
> ---
>
> **Question 4. Generalization**
>
> The CodeReviewer dataset is derived from diverse **public repositories on GitHub**. Therefore, our experiments on this benchmark serve as an indicator of the model's **generalization capabilities** within open-source environments. However, in practical deployment scenarios—particularly when transferring to distinct systems such as proprietary corporate environments—direct application of the model typically yields suboptimal results. **Fine-tuning with domain-specific data** remains essential, as coding standards and requirements vary significantly across organizations. Consequently, relying solely on the model's zero-shot generalization capability is insufficient to address these specific domain constraints.
>
> ---
>
> **Question 5. Human-centered quality evaluation**
>
> The metrics reported in this study serve as valid proxies for the helpfulness and correctness of code reviews. They can be interpreted as the statistical aggregation of **human judgments**, based on the premise that suggestions are typically accepted only when developers perceive them as beneficial. Since these interactions occur during actual development tasks, they capture **authentic reviewer-author dynamics** that are often difficult to replicate in controlled user studies. Nevertheless, we acknowledge that user studies provide a complementary and intuitive perspective on model effectiveness, and we will consider incorporating such human-centric evaluations in our future research.

---

### Official Review · Reviewer_fhKD · 2025-11-01

**Soundness:** 3
**Presentation:** 2
**Contribution:** 2
**Rating:** 4
**Confidence:** 5

**Summary:**

Good paper, but Limited Novelty

**Strengths:**

The paper presents Coder-R3, a model that learns to recognize, review, and repair code jointly. They implore supervised finetuning on top of the baseline Qwen2.5-Coder-7B.

++ The qualitative analysis via real-world deployment is interesting and shows the practical impact of this work.

**Weaknesses:**

- The paper could dive a bit more into the technical implementation. I am a bit unsure about the novelty in the implementation that led to higher performance.
- Missing evaluation on competitive models on the curated internal dataset.
- Missing comparison with the latest SWE-Agents, as I assume they are also capable of completing the three tasks separately.
- Missing citation to a relevant work: Review4Repair: Code Review Aided Automatic Program Repairing (Huq et al, 2021)

**Questions:**

n/a

---

> ### Author Response · Authors · 2025-12-01
>
> Thank you for your thoughtful review. We appreciate your recognition of our paper and address your concerns in detail below.
>
> ---
>
> **Question 1. Technical implementation**
>
> The novelty in the implementation that led to higher performance is primarily characterized by two key aspects:
>
> First, regarding the training methodology, we employed a **hybrid training strategy** that incorporates code **context information**. This approach effectively reduces model complexity while simultaneously enhancing performance.
>
> Second, within the actual production environment, we established an **iterative training framework** based on a novel **data flywheel** mechanism, designed to achieve continuous improvement in model efficacy. This framework is instrumental in establishing automated code review pipelines within corporate infrastructures.
>
> ---
>
> **Question 2. Evaluation on the internal dataset**
>
> The internal dataset experiments are intended to **supplement** the results on the CodeReviewer dataset. The purpose is to further demonstrate the effectiveness of our training method and to assess the contribution of each component through ablation studies. Since many competitive baselines rely on **non-released or proprietary implementations** that cannot be faithfully reproduced in our internal environment, so this comparison would be unreliable.
>
> ---
>
> **Question 3. Comparison with SWE-Agents**
>
> SWE-Agents are agentic systems that rely on multi-step execution and coordination across multiple model calls, whereas our work aims to enhance the intrinsic capability of a single model to perform the Recognize–Review–Repair tasks in one unified forward pass. Because agent-based pipelines and single-model training approaches are designed with different core objectives in mind, where the former focuses on coordinated multi-step orchestration and the latter focuses on strengthening **the intrinsic capabilities of the model itself**, a direct comparison would not offer a fair or informative evaluation.
>
> ---
>
> **Question 4. Missing citation**
>
> The paper you recommended is indeed a valuable contribution to the field of code review. We will cite this work in the revised version of our paper.

---

### Official Review · Reviewer_mXcQ · 2025-11-01

**Soundness:** 2
**Presentation:** 3
**Contribution:** 2
**Rating:** 4
**Confidence:** 3

**Summary:**

The paper addresses the fragmentation in LLM-based code review research by proposing a unified framework, Coder-R3, which models the review process as three steps: Recognize (detect defects), Review (generate explanations), and Repair (correct code). By fine-tuning a single large language model to handle all tasks, the authors explore joint training (generating all results at once) and hybrid training (handling each subtask separately). The paper systematically analyzes the impact of contextual inputs. A key contribution is the "data flywheel," collecting real-world human feedback to iteratively improve the model. Experiments show that Coder-R3 achieves state-of-the-art results on public and proprietary datasets, and its performance continues to improve in real-world deployment via the data flywheel mechanism.

**Strengths:**

The paper addresses automated code review deployment by integrating a unified model and iterative data strategy, directly targeting industrial software challenges.

The "Recognize-Review-Repair" (R3) framework captures the synergy between defect identification, explanation, and correction, avoiding fragmentation of isolated tasks.

Achieves SOTA on two key tasks in the CodeReviewer benchmark, confirming the reliability of the fine-tuning and data processing methods.

**Weaknesses:**

Coder-R3 underperforms dedicated models in generating review comments—a limitation noted but insufficiently analyzed. It remains unclear whether this stems from intrinsic trade-offs in multi-task objectives or limitations in training data/methodology. This analytical gap weakens the paper’s credibility.

The data flywheel, meanwhile, is painted in rosy hues. What about the messy realities—spam feedback, shifting code styles across teams, or the growing compute bill every time you retrain?

**Questions:**

Your results show that "hybrid training" (three invocations) outperforms "joint training" (one invocation). Could you elaborate on the practical advantages of your hybrid approach compared to using three distinct, potentially smaller, specialized models?

As mentioned above, the data flywheel relies on human feedback. How do you plan to address the challenge of noisy or low-quality feedback (e.g., reviewers lazily accepting all suggestions) as you scale the system? Are there automated quality checks in your "Data Filtering" step to mitigate this?

---

> ### Author Response · Authors · 2025-12-01
>
> Thank you for your thoughtful review. We appreciate your recognition of our paper and address your concerns in detail below.
>
> ---
>
> **Question 1. Hybrid training advantages**
>
> In the ablation study presented in **Table 2**, we evaluated both hybrid and multi-model setups with the same model size. The results indicate that the two settings yield **comparable performance**; notably, the unified model exhibits a distinct advantage in the 'Recognize' task. The performance of smaller models is substantially degraded, rendering them incomparable to our proposed solution.
>
> In the terms of same model size, the advantages of hybrid training are demonstrated in the following three aspects:
>
> **Reduced invocation cost and latency.** A single unified model eliminates the need to load and run three separate models. This could reduce GPU memory usage, end-to-end latency and the number of inference calls.
>
> **Lower maintenance and storage overhead.** Maintaining one model instead of three simplifies updating and monitoring. It also reduces storage requirements and engineering complexity on the deployment side.
>
> **Shared representations across tasks.** Even though the hybrid model leverages multiple prompt types, the underlying shared parameters allow cross-task knowledge transfer. This often leads to more robust representations and can improve generalization.
>
> ---
>
> **Question 2. Data flywheel**
>
> First, as detailed in **Section 3.3**, we subject the collected data to multiple rounds of Human-AI collaborative verification to enhance data quality. Instances identified as low-quality via LLM-based scoring are flagged and subsequently re-annotated by human experts.
>
> Second, all samples included in our internal dataset are drawn from real production code within the company. Reviewers are required to **carefully inspect** the code before accepting model recommendations and submitting the merged version. The quality of code reviews is also a component of employee working performance, which affects their ratings in the company. Therefore, the cases where low-quality data are introduced due to reviewer oversight are likely to be rare.
>
> Third, throughout the iterative training process, we consistently integrate newly acquired data into the cumulative dataset rather than training solely on the most recent samples. We observed that after a certain number of iterations, the model’s performance tends to **stabilize**, and we have not encountered any evidence of degradation or regression. This further suggests that the small amount of potentially low-quality data does not materially affect the model’s overall performance.

---

### Official Review · Reviewer_HdTi · 2025-11-01

**Soundness:** 2
**Presentation:** 2
**Contribution:** 1
**Rating:** 2
**Confidence:** 4

**Summary:**

This paper presents Coder-R3, a finetuned LLM that unifies defect detection, review generation, and bug fix a single model. In experiments and ablations, a hybrid invocation of the same model with code-diff context and full-snippet outputs consistently beats joint training and diff-style outputs, yielding clear best practices. The model reaches state-of-the-art on the CodeReviewer benchmark and shows steady real-world gains when deployed inside a company code-review system.

**Strengths:**

1. Coder-R3 shows substaintial performance gain on top of the origianl model (Qwen-2.5-Coder-7B) being fine-tuned.

2. A rather comprehensive collection of baselines are introduced and compared, making it informative for new readers in this area.

**Weaknesses:**

1. It seems that exsting works already have solution for unified role [1] .

2. The technical novelty and contribution of this paper is not clear, I fail to see any actual novel methodlogy

3. In Table 2, performance metrics of significant part of the baselines are omitted, and there's no reason to explain that.

4. There lacks sufficient details to the data construction, for example, what sources are considered, are they overlapping with the test cases you used? How does this dataset differs from the datasets from previous studies. What is the detailed statistics.

5. One important motivation seems to be the efficiency concern, but throughout this paper I cannot see any experiment justified this.

[1] CodeAgent: Autonomous Communicative Agents for Code Review, EMNLP 2024.

**Questions:**

N/A

---

> ### Author Response · Authors · 2025-12-01
>
> Thank you for your thoughtful review. We appreciate your recognition of our paper and address your concerns in detail below.
>
> ---
>
> **Question 1. Existing works**
>
> The methodology proposed in CodeAgent employs a LLM to simulate distinct roles, facilitating collaboration through a pipeline-based workflow. Although this approach utilizes a single general-purpose model (GPT), it operates on **an agent-based framework** where different roles necessitate repeated invocations of the LLM, thereby increasing overall system complexity.
>
> In contrast, the primary objective of our work is to enhance **the intrinsic capabilities of the model itself**. We aim to develop a unified model capable of simultaneously performing the Recognize-Review-Repair tasks, without relying on collaboration between multiple models or agents. Consequently, these two methodologies diverge significantly in their fundamental design and should not be conflated.
>
> ---
>
> **Question 2. Technical novelty**
>
> The innovation of our proposed method is primarily characterized by two key aspects:
>
> First, regarding the training methodology, we employed a **hybrid training strategy** that incorporates code **context information**. This approach effectively reduces model complexity while simultaneously enhancing performance.
>
> Second, within the actual production environment, we established an **iterative training framework** based on a novel **data flywheel** mechanism, designed to achieve continuous improvement in model efficacy. This framework is instrumental in establishing automated code review pipelines within corporate infrastructures.
>
> ---
>
> **Question 3. Table explanation**
>
> The metrics for all tasks in our experiments follow the experimental settings defined in CodeReviewer. However, a significant amount of subsequent related work has concentrated on **only one or two of these tasks** or **did not report certain metrics**. Consequently, we denoted these unavailable metrics with '-' in Table 2. We acknowledge that this notation was not explicitly clarified in the original text. We appreciate you pointing out this oversight, and we will include a clear explanation in the revised manuscript.
>
> ---
>
> **Question 4. Data construction details**
>
> Our dataset was constructed based on the company's internal code management platform, specific details are provided in Sections 3.2 and 5.1. The data originates from merge requests submitted by engineers during actual development workflows. The training and test sets were strictly partitioned chronologically to ensure there is **no data leakage or overlap**. Although our dataset is smaller in scale compared to existing public datasets, it offers **higher quality** and aligns more closely with **real-world industrial scenarios.**
>
> ---
>
> **Question 5. Efficiency concern**
>
> Hybrid training models are superior to traditional multi-model implementations in terms of efficiency and deployment. During the training phase, the multi-model approach necessitates the training of three distinct models, which significantly increases the **complexity of maintenance and updates**. In terms of inference, the unified framework requires only a single model invocation, thereby **reducing GPU memory consumption, minimizing the number of calls, and lowering end-to-end latency**.
>
> Consequently, given that the multi-model setup yielded no performance improvements while imposing greater deployment complexity, we did not deploy it in our production environment for comparative evaluation.

---

### Meta-Review · Area_Chair_n4Vb · 2026-01-12

**Summary:**

This paper proposes Coder-R3, a unified fine-tuned LLM that performs defect recognition, review generation, and code repair within a single framework, complemented by a hybrid training strategy and an industrial data flywheel. Reviewers broadly agreed that the paper is well-motivated from a practical and industrial perspective and that the empirical results, especially on CodeReviewer and in internal deployment, are solid. However, the decision to reject is driven by insufficient technical novelty, with multiple reviewers noting strong conceptual overlap with prior work on unified or agent-based code review systems. Additional concerns include fairness and completeness of experimental comparisons, heavy reliance on a single, imperfect public benchmark, limited justification of evaluation metrics, and reproducibility and generalization limitations stemming from proprietary data and deployment settings. While the system engineering and deployment narrative is compelling, the overall contribution falls short in methodological and novelty.

**Reviewer Concerns:**

The rebuttal successfully clarified several implementation and system-level questions, including the rationale for hybrid vs. joint training, the structure and maintenance benefits of a unified model, mitigation of noisy human feedback in the data flywheel, and reasons for not comparing directly with agent-based systems or closed-source models. The authors also acknowledged missing citations and promised releases of checkpoints and scripts, partially addressing reproducibility concerns.

However, key issues remain unresolved. Most reviewers remained unconvinced about the core technical novelty, viewing hybrid training and the R3 formulation as incremental refinements rather than new methodology. Concerns about evaluation fairness, benchmark dependence, limited use of code-appropriate metrics, and the lack of public validation of the industrial flywheel persist. The rebuttal largely defends design choices rather than providing new evidence that would change these assessments.

**Reviewer Scores:**

Reviewer HdTi would likely not change their score, as concerns about novelty, missing experimental efficiency, and dataset transparency remain largely unaddressed. Reviewer mXcQ might remain at a borderline reject, as questions about review-quality degradation, noisy feedback, and practical trade-offs were discussed but not empirically resolved. Reviewer fhKD would likely maintain their score, since novelty and missing competitive comparisons remain open. Reviewer 7wdc might slightly soften their stance but would likely stay below the acceptance threshold, given persistent concerns about evaluation fairness, metrics, and external validity. Reviewer UVXn would likely not change their score, as issues of limited novelty, lack of theory, reproducibility, and generalization remain.

---

### Decision · Program_Chairs · 2026-01-26

Reject